



# The Holton-Tan mechanism under stratospheric aerosol intervention

Khalil Karami[1], Rolando Garcia[2], Christoph Jacobi[1], Jadwiga H. Richter[2], and Simone Tilmes[2]

[1]Institute for Meteorology, Leipzig University, Leipzig, Germany
[2]National Center for Atmospheric Research, Boulder, CO, USA

**Correspondence:** Khalil Karami (khalil.karami@uni-leipzig.de)

**Abstract.** The teleconnection between the Quasi-Biennial Oscillation (QBO) and the Arctic stratospheric polar vortex, or the Holton-Tan (HT) relationship may change in a warmer climate or one with stratospheric aerosol intervention (SAI) as compared to present day climate (PDC). Our results from an Earth system model indicate that, under both global warming (based on RCP8.5 emission scenario) and SAI scenarios, the HT relationship weakens, although it is closer to PDC under
SAI than under the RCP8.5 scenario. Such weakening of the HT relationship is more pronounced in early winter (Nov-Dec) compared to the mid-late winter period (Jan-Feb). While the high-latitude responses of temperature to the QBO anomalies are statistically significant under PDC, the responses are not statistically significant in the RCP8.5 and SAI scenarios. While the weakening of the HT relationship under RCP8.5 scenario is likely due to the weaker QBO wind amplitudes at the equator, another physical mechanism must be responsible for the weaker HT relationship under SAI scenario, since the amplitude of the
QBO wind is comparable to the PDC. The strength of the polar vortex does not change under the RCP8.5 scenario compared to PDC, but it becomes stronger under SAI; we attribute the weakening of the HT relationship under SAI to such stronger polar vortex. In general, the changes in the HT relationship cannot be solely explained by changes to the critical line; the changes in the residual circulation (particularly due to the gravity wave contributions) are important too in explaining the changes in the HT relationship under RCP8.5 and SAI scenarios.

## 1  Introduction

In addition to mitigation and adaption efforts, the changing climate demands consideration and research on other responses as it is increasingly becoming evident that the current trajectory of the greenhouse gas emissions and increases in the global mean temperature inevitably leads to climate that is dangerous. There exists a chance of a 5% and 1% achieving the Paris Agreement's goal of limiting the global mean surface temperature increase to below 2 °C and 1.5°C, respectively (Raftery et
al. , 2017). Hence, Solar Radiation Management (SRM), which attempts to intentionally modify the Earth's radiation budget (IPCC , 2018), focuses on the near-term risk reduction (associated with a dangerous climate change) to a manageable level that cannot be achieved by emission cuts alone (Crutzen , 2006; Rasch et al. , 2008). Stratospheric aerosol intervention (SAI), which has received particular attention, is a proposed method aiming reflecting of a small percentage of incoming shortwave





radiation back to space, thus mimicking the cooling effect of volcanic eruptions in reducing surface temperatures (Kravitz et
al. , 2013; Tilmes et al. , 2018a).

SAI is thought to be effective at moderating key climate hazards (Irvine et al. , 2019), but does so imperfectly and presents
its own risks including ocean acidification (Lauvset et al. , 2017), ozone losses from sulfur injections (Tilmes et al. , 2008),
and unequal and nonuniform regional compensation in temperature and precipitation distributions (Robock , 2008; Kravitz
et al. , 2014). Thus, it remains unclear whether the risks of SAI exceed or fall short of the risks of breaking the 2°C target
(Parker and Irvine , 2018; Rahman et al. , 2018) and such risks and benefits needs to be evaluated for all parts of the Earth
system. Although SAI should not be viewed as a tool to achieve any arbitrary set of climate objectives, Kravitz et al. (2017),
using a state-of-the-art climate model demonstrated that strategically performed SAI is effective to meet multiple simultaneous
temperature objectives in the presence of uncertainty. To maintain three surface temperature features (global mean, equator-to-
pole gradient, and interhemispheric gradients) close to their state in 2020 under the RCP8.5 emission scenario (Riahi et al. ,
2011) from 2020 to 2099, they used a feedback mechanism in which the rate of sulfur dioxide injection is adjusted in every year
of the simulation at four latitudes. Because the above-mentioned objectives are defined in terms of annual mean temperature,
the required amount of sulfur dioxide injection leads to seasonally different responses in temperature as well as hydrological
cycle compared to year 2020 (Kravitz et al. , 2017).

SAI changes not only the surface climate but also changes the stratospheric climate primarily due to stratospheric heating
from injected aerosols. While in SAI the injected sulfur dioxide aerosols cool the Earth's surface, they heat the tropical lower
stratosphere and cool the wintertime Northern Hemisphere (NH) polar stratosphere, which leads to the strengthening of the
Arctic polar vortex (Stenchikov et al. , 2002; Tilmes et al. , 2009; Ferraro et al. , 2011), and a near elimination or reduction
of Sudden Stratospheric Warmings (Ferraro et al. , 2015; Richter et al. , 2018). Such temperature changes can potentially
influence stratospheric chemistry, including concentrations of ozone, particularly in the middle and high-latitudes (Tilmes et
al. , 2009). Previous research indicates that the latitude of the injection of aerosols has a great impact on the periods of the
quasi-biennial oscillation (QBO). For example, Richter et al. (2018), using the Community Earth System Model, version 1,
with the Whole Atmosphere Community Climate Model CESM1(WACCM) as its atmospheric component, showed that the
QBO period greatly lengthens (to about 3.5 years compared to 24 months in their control simulation) in response to equatorial
injection of sulfur dioxide at about 5 km above the tropopause at a constant emission rate of 12 Tg/year. On the other hand, they
show that non-equatorial SO2 injection (at a single point) shortens the QBO period to about 12-17 months. The result of further
research show that, when such aerosols injections are located simultaneously on both sides of the equator, the QBO period is
less impacted (Kravitz et al. , 2019). The unaffected QBO, among other smaller side effects, motivated the Stratospheric
Aerosol Geoengineering Large Ensemble Project (GLENS). GLENS is further described in section 2.1.

While the QBO is a tropical stratospheric phenomenon, it impacts the tropical, subtropical and extra-tropical troposphere
through a variety of mechanisms. In the tropics, the tropical deep convection as well as Madden Julian Oscillation (MJO)-
like convective activity are significantly modulated by the QBO (Son et al. , 2017). In the subtropics, QBO influences the
strength and location of the jet streams, storm tracks (Wang et al. , 2018), and tropospheric eddies (Inoue et al. , 2011). The
structure of the QBO, e.g., its vertical extent (Andrews et al. , 2019), its meridional extent (Hansen et al. , 2013), and its



extension into the lower stratosphere (Collimore et al. , 2003) influences its teleconnections with other components of the

large-scale atmospheric circulation. In addition, the QBO influences the strength of the wintertime stratospheric polar vortex. More specifically, the polar vortex becomes weaker and warmer during its easterly phase (QBOe) and stronger and colder during its westerly phase (QBOw), a mechanism known as the Holton and Tan (1980) relationship (HT relationship hereafter). Holton and Tan (1980, 1982) referred to the work of Tung et al. (1979), who argued that the critical wind (zero wind) surfaces tend to reflect small amplitude planetary waves toward the mid and higher latitudes. They hypothesized that under QBOe

the subtropical zero wind line is near the winter subtropics, and this narrows the width of the extratropical waveguide for the upward wave propagation, which results in a decelerated/warmer polar vortex. Conversely, under QBOw, the zero-wind line shifts toward the equator, which makes it easier for the waves to propagate to lower latitudes, and this results in a less disturbed/colder polar vortex. Despite the apparent observational support for the HT relationship, the critical line mechanism depends on several assumptions that are not fully met in the real atmosphere. It is worthwhile to mention that the strength of

the HT relationship is transient and changes over time. For example, Lu et al. (2008) reported that the HT relationship was robust during 1958-1976 and weak during 1977-1997. This is because the QBO signal accounts for only a fraction of the polar vortex variance and other factors such as the El Niño-Southern Oscillation (ENSO), volcanic eruptions and the 11 year solar cycle can also disrupt the HT relationship (Garfinkel et al. , 2008; Wei et al. , 2007; Stenchikov et al. , 2004; Labitzke et al. , 1988).

In addition to the critical line mechanism, another explanation for the observed HT relationship is the QBO-induced meridional circulation. Above and below the peak QBO winds, a thermal balance exists between the vertical shear of the QBO wind and the QBO-induced temperature anomalies (Baldwin et al. , 2001). Such a temperature anomaly pattern must include a residual mean meridional circulation that maintains the dynamically forced QBO-induced temperature anomalies against radiative relaxation (Garfinkel et al. , 2012; Plumb and Bell , 1982). It is suggested that the QBO-induced meridional circulation at low

latitudes may alter the background flow that affects the planetary wave propagation and therefore results in the extratropical response (Kodera et al. , 1991; Ruzmaikin et al. , 2005).

Because of the high predictability potential of the QBO itself, it is suggested that the above-mentioned QBO teleconnection could be used to improve the forecasting skill of seasonal to subseasonal climate predictions. Robertson et al. (2020) provided that models are able to (a) realistically represent the HT relationship and (b) skillfully represent the dynamical coupling between

the stratosphere and troposphere (Butler et al. , 2019). With a warming climate, however, the HT relationship may have another mechanism and pathway, contrasting to the present-day climate. While most Coupled Model Intercomparison Project 6 (CMIP6) models underestimate the HT relationship in the present-day climate (Elsbury et al. , 2021) , there exist inconsistent QBO period responses among models (shortening by 8 months in some models and lengthening by up to 13 months in others) in response to doubled $CO_2$ simulations (Richter et al. , 2022). The low confidence level for the future projection of the QBO

in modelling studies is partly attributed to the different choice of non-orographic gravity wave parameterization employed in different models, and that their properties such as launch level, spectral shape, physical link to the sources and propagation scheme are different (Schirber et al. , 2015). In addition, most models from the CMIP5/6 project show an enhanced surface response to the QBO due to an enhanced HT relationship under a warming climate (Rao et al. , 2020). Such strengthening of





the HT relationship is reported even when the amplitude of the QBO weakens in the projections compared to the historical periods, which highlights the importance of the nonlinear relationship between the QBO intensity and its teleconnections (Rao et al. , 2020).

While possible changes in stratospheric dynamics (including the QBO period) due to SAI have been considered by several studies (Tilmes et al. , 2018a; Richter et al. , 2017, 2018), the possible changes in the HT relationship under SAI have not been considered in detail. Consequently, this study has the following objectives: (1) to analyse to what extent global warming and SAI modulate the HT relationship in the stratospheric extratropical response to the QBO phases, and (2) to gain further insight into the physical mechanism involved in the possible modulation of HT relationship under global warming and SAI scenarios. The rest of the paper is organised as follows: Section 2 describes the GLENS simulation as well as analysis methods. By using various wave-mean flow interaction diagnostics, in section 3 we analyse the modulation of the HT relationship to warming and SAI scenarios, and finally in section 4, we summarize the major findings.

## 2    Data and Method

### 2.1    GLENS Simulations

We use here the GLENS project (Tilmes et al. , 2018b) to assess the potential impacts of SAI and global warming on the HT mechanism. GLENS was carried out with the Community Earth System Model version 1, CESM1, which has fully coupled atmospheric, ocean, land, and sea ice components. The CESM1 set-up for GLENS used the Whole Atmosphere Community Model (WACCM) as its atmospheric component, with a horizontal resolution of $0.9°$ in latitude by $1.25°$ in longitude with a relatively coarse vertical resolution of 70 layers between surface and model top at 140 km. The simulation includes fully interactive middle atmosphere chemistry with 95 solution species, 2 invariant species, 91 photolysis reactions, and 207 other reactions (Mills et al. , 2017). The GLENS project consists of a 20-member ensemble of stratospheric sulfur aerosol injection simulations (hereby referred to as SAI simulation), covering the period of 2020-2099. Additionally, a 20-member ensemble of control simulations of RCP8.5 simulations (Riahi et al. , 2011), over a reference period between 2010-2030 (referred to as present-day climate simulation, or PDC) was produced. Three of the RCP8.5 members were continued until 2097 (and hereafter are referred to as RCP8.5 simulation). In the SAI simulation, sulfur dioxide is injected at 30°N, 30°S, 15°N and 15°S in the stratosphere (5 km above the tropopause), in order to keep the global surface temperature, interhemispheric and equator-to-pole temperature gradients at 2020 conditions by applying the feedback-control algorithm to control the amount of injection for each ensemble members separately (Kravitz et al. , 2017).

Here all 20 ensemble members of the GLENS simulations are used to represent the PDC (2010-2029) and SAI (2060-2079) simulations. The 3 available ensemble members from the RCP8.5 (2060-2079) simulation are used. It is worthwhile to mention that in CESM1(WACCM) both orographic and non-orographic (frontal and convectively generated) gravity waves (GWs) are parameterized based on the GWs source specification of Richter et al.  (2010), with the GWs propagation scheme of (Lindzen et al. , 1981). By increasing the efficiency of convectively generated GWs and increased horizontal resolution compared to (Mills et al. , 2016), CESM1(WACCM) generates an internal QBO (Mills et al. , 2017). The mean period of the simulated QBO ranges





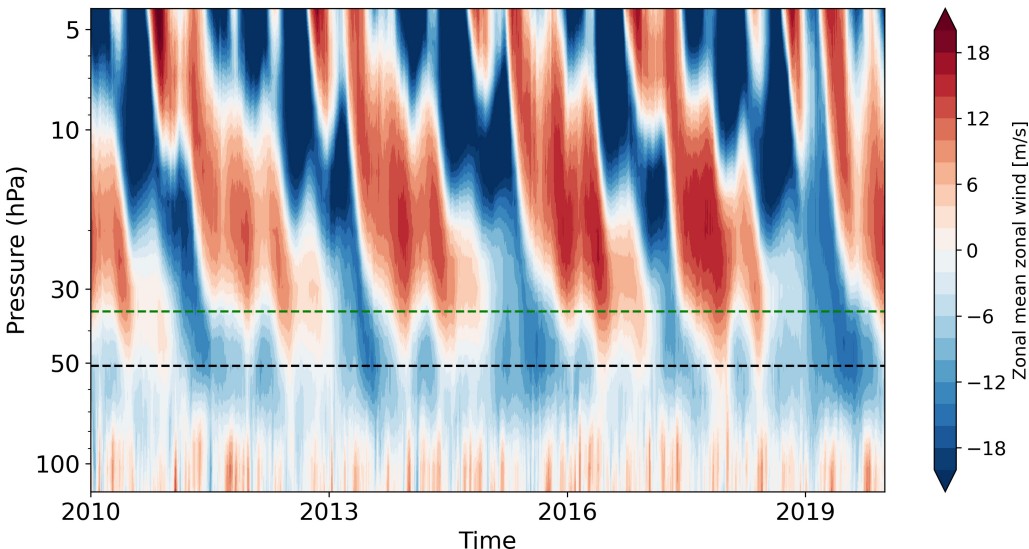

**Figure 1.** The equatorial (5°S–5°N) zonal mean zonal winds from 2010-2020 based on the present-day climate of the GLENS simulation.

from 23 to 27 months for ensemble members. In observations, the mean QBO period is about 28 months and ranges between 20 and 34 months. The amplitude of the simulated westerly QBO phase is comparable to observations. However, the amplitude of the easterly phase is weaker than in observations (Figure 4a and 4b of (Mills et al. , 2017)). The simulated QBO is somewhat

deficient in the lower stratosphere as the westerly phases does not reach down to about 100 hPa as in observations, but it is confined to above approximately 50 hPa (see Fig 1). Richter et al. (2022) reported that most CMIP6 models systematically underestimate the amplitude of the lower stratospheric QBO.

Despite the well established relationship between the QBO phases and the strength of the Northern Hemisphere (NH) stratospheric polar vortex, it still remains unclear which vertical levels of the QBO exert the strongest influence on the polar

vortex. In particular it remains an open question that if a deep layer of westerly or easterly equatorial winds influences the high-latitude polar vortex, why a QBO definition of 50 hPa based on the equatorial zonal mean zonal wind (Holton and Tan , 1980; Garfinkel et al. , 2012) is a better choice than other levels (Anstey et al. , 2014). Indexing the QBO at 50 hPa is not feasible in GLENS simulation due to the weak amplitude of the simulated westerlies there (see Fig. 1). Instead, the closest higher level (level=42 which is at about 35 hPa) is selected to index the QBO where both easterlies and westerlies have large

enough amplitudes to detect the QBO phases. The QBO is indexed using the zonally averaged zonal wind $\overline{U}$ between 5°S–5°N, $\overline{U}(35hPa) > +2ms^{-1}$ as QBOw and $\overline{U}(35hPa) < -2ms^{-1}$ as QBOe. The composite differences of variables are expressed as QBOe minus QBOw. As the strength of the QBO modulation of the planetary waves changes from early winter to late winter (Holton and Tan , 1982; Hu and Tung , 2002), therefore we study the early winter (Nov-Dec) and mid-late winter (Jan-Feb) responses separately. We use Student's t-test at 95% significance to determine whether two datasets are significantly different.

Ideally, the GLENS's historical simulation (1980-2010) might be used to evaluate the model performance in simulating the HT effect (by comparing to ERA5 reanalysis during similar period). Unfortunately, the historical simulation is a single member





**Figure 2.** Scatter plot of the strength of the stratospheric polar vortex at 10 hPa, versus the strength of the equatorial zonal mean zonal wind in ERA5 (QBO defined at 50 hPa) and GLENS (QBO defined at 35hPa) simulations. Text in each panel gives the results of an ordinary least squares regression through the points.





and given a large variability of the strength of the polar vortex, such evaluation is not reliable (figures not shown). Instead, we use ERA5 reanalysis (Hersbach et al. , 2020) for the period 1980-2019 to evaluate the PDC (2010-2029, consisting of 20 ensemble members) for the representation of the QBO modulation of the stratospheric polar vortex in the GLENS simulations.

## 2.2 Transformed Eulerian-mean equation and waveguide metrics

We employ a number of diagnostics, including the Eliassen-Palm (EP) flux and its divergence, the quasi-geostrophic refractive index and the mean meridional circulation. The dynamical processes can be described based on the transformed Eulerian-mean (TEM) momentum equation, which connects the changes of the circulation to wave forcing (Andrews et al. , 1987). It can be expressed as:

$$\frac{\partial \overline{u}}{\partial t} = -(\frac{1}{a cos\phi}(\overline{u}cos\phi)_{\phi} - f)\overline{v}^* - \overline{u}_z\overline{w}^* + \frac{1}{\rho_0 a cos\phi}\nabla \cdot \boldsymbol{F} + \overline{X} \tag{1}$$

where $u$, $v$ and $w$ are Eulerian zonal, meridional, and vertical winds, $\phi$ is the latitude, $a$ is the Earth's radius, $f$ is the Coriolis parameter, $\rho_0 = \rho_s exp(\frac{-z}{H})$ is the standard density in log-pressure coordinates, where $\rho_s$ is the reference density at $1000hPa$ and $H$ is the scale height (7 km). Subscripts denote derivatives with respect to the given parameter. The first and second term on the right-hand side of Eq.1 are the acceleration associated with the TEM meridional circulation $(\overline{v}^*, \overline{w}^*)$, where

$$\overline{v}^* = \overline{v} - \frac{1}{\rho_0}(\rho_0 \frac{\overline{v'\theta'}}{\overline{\theta}_z})_z \tag{2}$$

$$\overline{w}^* = \overline{w} + \frac{1}{a cos\phi}(cos\phi \frac{\overline{v'\theta'}}{\overline{\theta}_z})_{\phi} \tag{3}$$

Here $\overline{v}$ and $\overline{w}$ are the components of the Eulerian mean meridional circulation and $\theta$ is the potential temperature.

The third term on the RHS of Eq. (1) is the divergence of the quasi-geostrophic version of the EP flux, which is proportional to the eddy heat and momentum fluxes. The EP flux indicates the strength and direction of large-scale wave propagation. The divergence of the EP flux, which represents the zonal forcing on the mean zonal flow from the large-scale resolved waves, is defined as:

$$\nabla \cdot \boldsymbol{F} = \frac{1}{a cos\phi}[F^{(\phi)}cos\phi]_{\phi} + F^{(z)}_z \tag{4}$$

where the vertical and meridional components of the EP-flux are:

$$F^{(z)} = \rho_0 a cos\phi((f - \frac{(\overline{u}cos\phi)_{\phi}}{a cos\phi})\frac{\overline{v'\theta'}}{\overline{\theta}_z} - \overline{w'u'}) \tag{5}$$

and

$$F^{(\phi)} = \rho_0 a cos\phi(\frac{\overline{v'\theta'}}{\overline{\theta}_z}\overline{u}_z - \overline{v'u'}) \tag{6}$$







**Figure 3.** Latitude-pressure cross section of the composite differences between QBOe and QBOw (QBOe-QBOw) for November-December (left column) and January-December (right column) of the zonal-mean zonal wind. The stippled areas indicate regions where the changes are not statistically significant at 95% level according to the t test. The green contours represent the zonal mean zonal winds of the respective dataset/simulation.





The last term in Eq. (1) is the forcing by dissipation and other unresolved processes such as gravity wave drags. The stream function of the residual mean meridional circulation can be estimated by the vertical integration of $\overline{v}^*$:

$$\Psi_{direct}(\phi,z) = -\int_z^{top} \rho_0 cos\phi\overline{v}^* dz \tag{7}$$

Here, we adapt the methodology introduced by Okamoto et al. (2011) to calculated the relative contributions of the resolved Rossby waves versus unresolved zonal force (e.g., gravity wave drag) in driving the mean residual circulation. Their method is based on the "downward control principle (DCP)" proposed by Haynes et al. (1991) where the DCP suggests that the meridional residual flow in the middle atmosphere is largely driven the dissipating waves in the mid-latitudes and the vertical flows are then induced in the tropics and lower latitudes by mass conservation. The DCP also states that in the steady-state case and neglecting the vertical advection by the residual mean flow, the extratropical meridional mass flow at a given pressure level is determined solely by the sum of all zonal forces above the pressure level. Therefore,

$$-\hat{f}\overline{v}^* = \frac{1}{\rho_0 acos\phi}\nabla\cdot\boldsymbol{F} + \overline{X} = \overline{F} \tag{8}$$

where

$$\hat{f} \equiv f - \frac{1}{acos\phi}\frac{\partial(\overline{u}cos\phi)}{\partial\phi} = 2\Omega sin\phi - \frac{1}{acos\phi}\frac{\partial(\overline{u}cos\phi)}{\partial\phi} \tag{9}$$

In pressure coordinates, we have:

$$\Psi_{epfd}(\phi,p) = \frac{cos\phi}{g}\int_p^0 \frac{\overline{F}}{\hat{f}}dp' \tag{10}$$

and

$$\Psi_{gwd} = \Psi_{direct} - \Psi_{epfd} \tag{11}$$

Here, the subscript "direct" is used for the results calculated by Eq. (7), which accounts for both resolved and unresolved wave forcings, the subscripts "epfd" and "gwd" represent the contribution of the zonal forcing by resolved waves and parameterized gravity waves, respectively, and $g$ is the gravitational acceleration. For the wave guide metric, we use the method of Karami et al. (2016). The probability of favorable propagation conditions for Rossby waves is based on the frequency distribution of days with positive vertical wavenumber squared, which was originally introduced by Matsuno (1970) as a diagnostic tool for studying the influence of the background zonal flow on the large-scale planetary wave (PW) propagation. According to linear wave theory, the PWs tend to propagate to the regions where the vertical wavenumber squared is positive and avoid regions with small or negative values of this quantity. Here we use a two-dimensional (depending on the zonal and meridional wavenumbers) formulation of the vertical wavenumbers (Sun and Li , 2012; Sun et al. , 2014)

$$m_{k,l}^2(\phi,z) = (\frac{N^2}{f^2cos^2\phi})[\frac{\overline{q_\phi}}{\overline{u}} - (\frac{k}{a})^2(\frac{\pi l}{2a})^2 - (\frac{f cos\phi}{2NH})^2] \tag{12}$$





where

$$\overline{q_\phi} = cos\phi[\frac{2\Omega}{a}cos\phi - \frac{1}{a^2}\frac{\partial}{\partial\phi}[\frac{\frac{\partial}{\partial\phi}(\overline{u}cos\phi)}{cos\phi}] - \frac{f^2}{\rho_0}[\frac{\partial}{\partial z}\frac{(\rho_0\frac{\partial}{\partial z}\overline{u})}{N^2}]] \quad (13)$$

is the meridional gradient of the zonally-averaged potential vorticity (Andrews et al. , 1987). Here $k$, $l$, $N^2$ and $\omega$ are the zonal and meridional wavenumbers, buoyancy frequency and the Earth's angular rotation frequency, respectively. Here we only consider the stationary planetary waves with zero phase speed.

## 3 Results

### 3.1 Influence of the Equatorial QBO on the Extratropical Circulation

Scatter plots of the QBO zonal-mean winds versus the stratospheric polar vortex wind at 10 hPa and 60°N are shown in Fig.2. For ERA5, the equatorial zonal-mean wind,$\overline{u}$, is defined at 50 hPa and for the GLENS simulation at 35 hPa (see section 2) and therefore slightly different responses between the two datasets are expected (note that the period covered by ERA5 and PDC are also different). Both ERA5 and the PDC simulation of the GLENS project suggest that the strength of the HT relationship is

stronger in Jan-Feb compared to the Nov-Dec period. This also holds true for the RCP8.5 and SAI simulations. The strength of the HT relationship in SAI is weaker compared to the PDC of GLENS simulation. While in Nov-Dec the HT relationship is also weaker in the RCP8.5 compared to the PDC, such relationship is stronger in Jan-Feb period. Richter et al. (2018) showed that the period and the amplitude of the QBO are affected by both global warming and SAI. They showed that the mean QBO period decreases to 14 months by the end of century (2080-2099) under the RCP8.5 scenario compared to 24 months in 1980-1999.

They attributed the shortening of the QBO period under the RCP8.5 scenario of the GLENS simulation to an enhancement of the amplitude of tropical convective heating that increases the gravity wave momentum flux entering the stratosphere. In the SAI simulation, the mean period also decreases, to about 21 months in 2080-2099, but it remains closer to what is simulated than in the RCP8.5 simulation. The reason is that in the SAI simulation the sea surface temperatures, and hence the strength of convection, remain close to 2020 levels and therefore the gravity wave momentum entering the stratosphere remains at

those levels, such that the QBO period does not change significantly. As shown in Fig. 2g and Fig. 2h, the easterlies under SAI dominate the QBOe phase compared to the PDC, and westerlies weaken, which is more pronounced in Nov-Dec. This is associated with the change in the climatology of the zonal mean zonal wind towards a stronger easterly state under the SAI scenario (Tilmes et al. , 2018a; Richter et al. , 2017).

Figures 3 and 4 show the QBO composite differences (QBOe-QBOw) of zonal mean zonal wind and temperature averages,

respectively. In the equatorial region, in both ERA5 and GLENS, the response of $\overline{u}$ is most significant at the pressure levels where the phase of the QBO was defined for each dataset. In the high-latitude stratosphere, the weakening of $\overline{u}$ and correspondingly warmer polar region marks the HT effect. While the PDC of the GLENS simulation underestimate the HT relationship compared to ERA5 in Jan-Feb, a stronger HT relationship is found in Nov-Dec. Nevertheless, despite the differences between the PDC of the GLENS simulation and ERA5 reanalysis, the pattern of the HT relationship is reasonably represented in the

PDC of the GLENS simulations. Both ERA5 and GLENS PDC agree on the larger areal extent of the HT effect in the early





**Figure 4.** The same as Fig. 3 but for temperature. The stippled areas indicate regions where the changes are statistically significant at 95% level according to the t test.

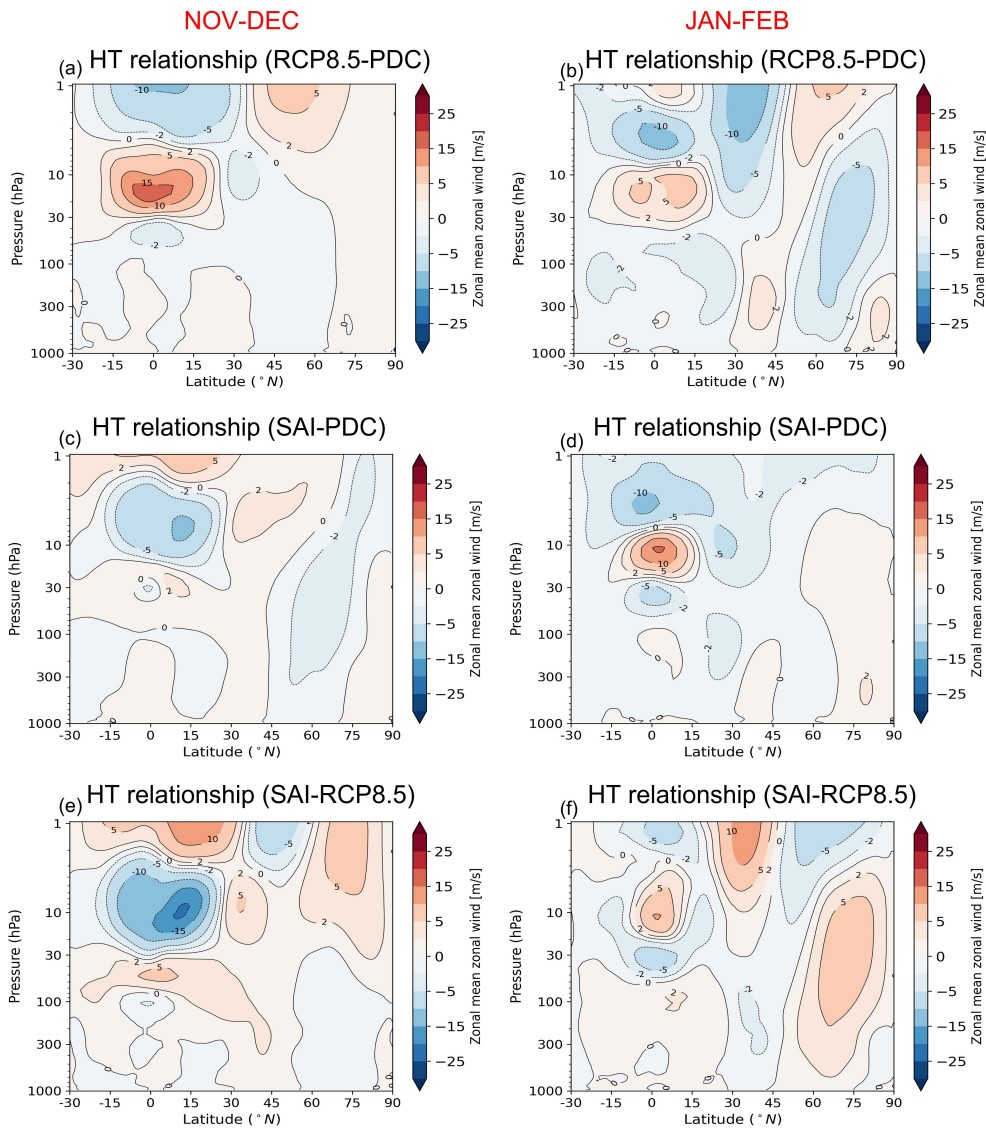

**Figure 5.** Changes in the HT relationship (for zonal mean zonal wind) under SAI and global warming (RCP8.5) scenarios compared to the PDC.

winter (Nov-Dec). The well-known, three-vertical-cell structure of the temperature anomalies is evident both at the equator and in the subtropics in all datasets. The structure is geostrophically consistent with the wind structure (Fig. 3), and is maintained by adiabatic cooling and warming associated with the secondary meridional circulation of the QBO (Plumb and Bell , 1982); this will be discussed further in the next section.

To examine the changes in the HT relationship between the SAI and RCP8.5 scenarios compared to the PDC simulation, Fig. 5 and Fig. 6 show, respectively, the composite differences of the zonal mean zonal wind and temperature responses due to the



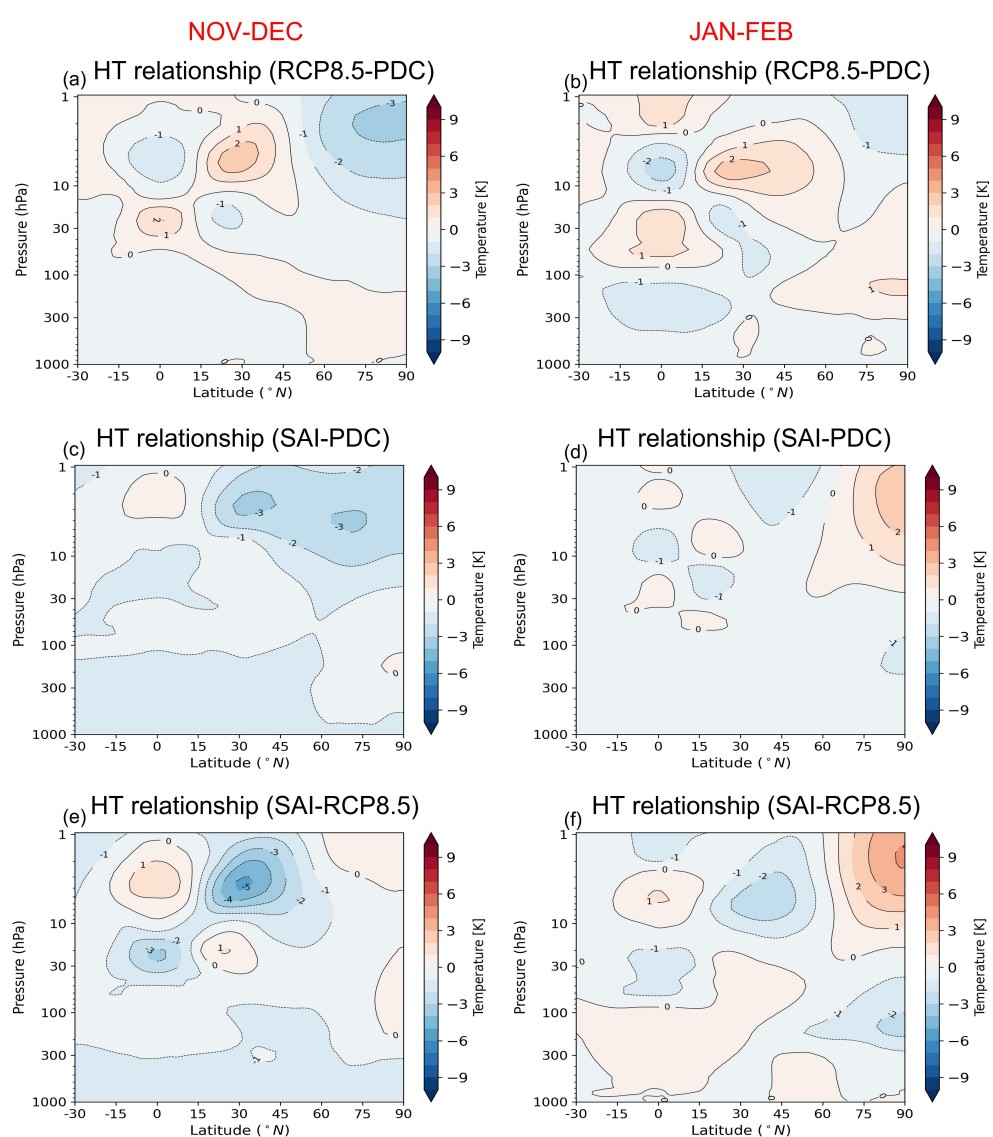

**Figure 6.** The same as Fig. 5 but for the zonal mean temperature.





HT relationship (presented in Fig 3 and Fig. 4). In RCP8.5, the QBO central anomaly at 35 hPa is not only weaker than in PDC but its vertical and meridional extent also covers a smaller area. In the SAI simulation the QBO central anomaly also weakens slightly compared to PDC, but it remains closer in both magnitude and vertical extent of the anomalies to what is simulated under PDC than in the RCP8.5 simulation. The positive/negative zonal mean zonal wind (Fig. 5a-5d) and negative/positive temperature (Fig. 6a-6d) anomalies in the high-latitude stratosphere indicate a weaker/enhanced HT relationship under either RCP8.5 or SAI scenarios compared to the PDC. While the high-latitude responses of temperature to the QBO anomalies are statistically significant in PDC, they are not significant under both RCP8.5 and SAI scenarios (Fig. 4). The weakened HT relationship (indicated either by the smaller responses or by the statistically non-significant responses) under SAI scenario cannot be attributed to the changes in the QBO anomalies in the equatorial region, as it largely resembles the QBO anomalies under the PDC simulation. However, under a warming climate (RCP8.5 simulation) it is more likely that the changes of the HT relationship are due to the smaller magnitude of the QBO anomalies in the equatorial region (Fig. 3e and 3f) compared to the SAI.

## 4   EP flux, refractive index and residual circulation responses

Figure 7 shows the composite QBOe-QBOw differences of the EP flux, its divergence, the zonal mean zonal wind response, and the location of the zero wind line for the different QBO phases. As expected from the HT hypothesis, we found that under QBOe the climatologically equatorward component of the EP flux is weaker than under QBOw, and hence the composite difference shows that under different climate change scenarios the resolved waves are always directed towards the high latitudes. This is consistent with the traditional HT hypothesis (Holton and Tan , 1980) stating that under QBOe, the quasi-stationary waves cannot propagate in easterly winds and therefore the effective waveguide for the planetary waves is narrower compared to the QBOw case. This is shown in Fig. 8 as the negative anomaly of the favorable propagation condition for Rossby waves centered at 35 hPa. The narrower waveguide may be viewed as inhibiting equatorward planetary wave propagation (or enhancing refraction to high latitudes), and also inhibiting upward wave propagation (Naoe and Shibata , 2010). While the zero-wind line is on the winter side of the equator (20-30 km) for QBOe under the PDC and SAI scenarios, the distance between the location of the zero-wind line is smaller for the different QBO phases under RCP8.5 (particularly in Jan-Feb). This is also reflected in Fig. 8 where the anomalies of the favorable propagation condition for Rossby waves are meridionally limited and also weaker under RCP8.5 compared to the SAI and PDC simulations particularly in Jan-Feb. This is also expected as the zonal mean zonal wind anomalies (Fig. 3 show similar patterns. While the weaker equatorward EP flux component in QBOe-QBOw composites are a persistent feature in all scenarios and periods, the vertical EP flux component shows inconsistent responses. In general, in early winter (Nov-Dec) an enhanced upward EP flux is found in the high-latitude stratosphere in all scenarios. On the other hand, in mid-late winter (Jan-Feb) a reduction of the upward EP flux is found in all scenarios. The direction of the downward EP flux anomalies in the high-latitude stratosphere are consistent with the HT hypothesis (Holton and Tan , 1980) in Jan-Feb period. On the contrary, the upward EP flux anomalies in the high-latitude stratosphere are inconsistent with the HT hypothesis for Nov-Dec in all scenarios.

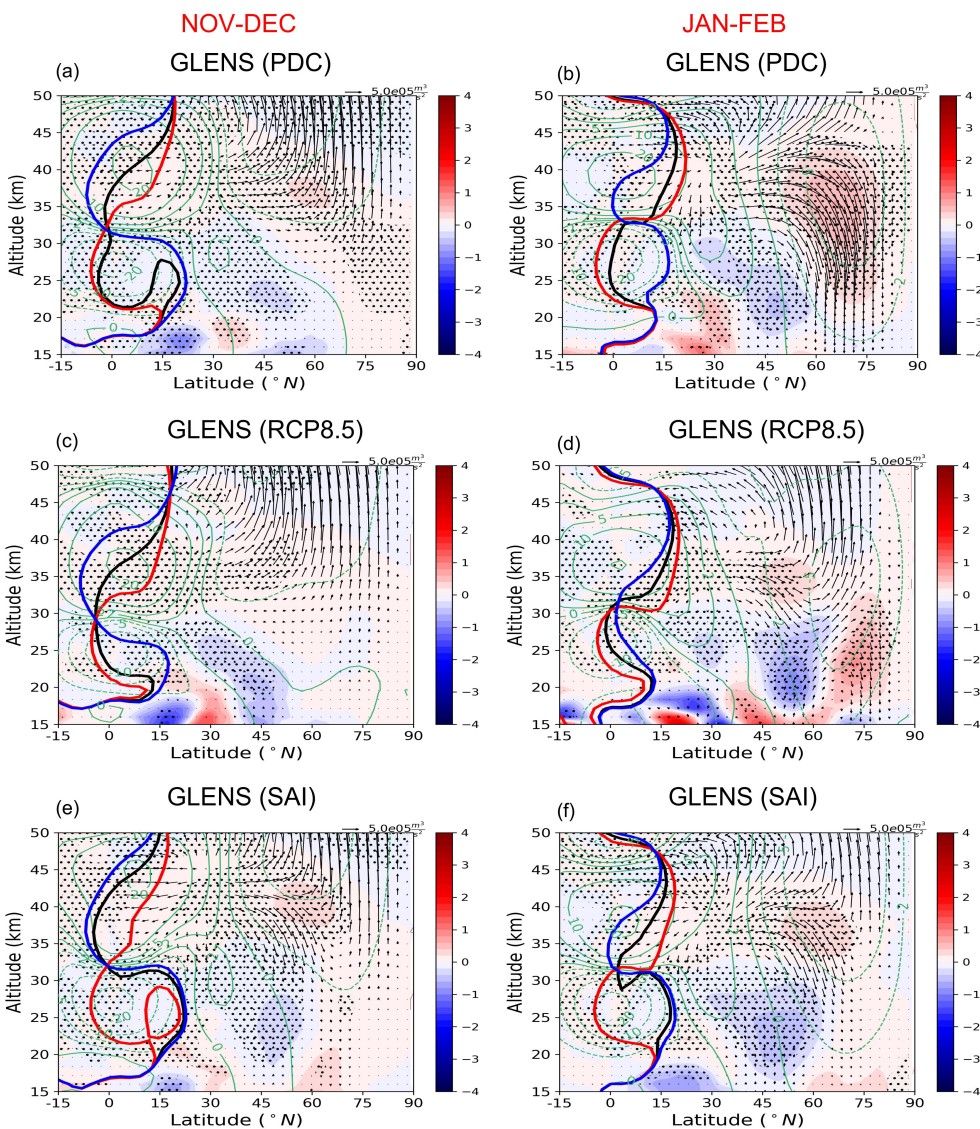

**Figure 7.** Latitude-height cross section of the QBOe-QBOw composite differences of the EP flux (arrows) and its divergence (color shaded in m/s/day). The green contours are the zonal mean zonal wind responses and the solid red, blue and black lines are the location of the zero-wind line for the QBOw, QBOe, and climatology, respectively. The stippled areas indicate regions where the changes of the EP flux divergence are statistically significant at 95% level according to the t test.

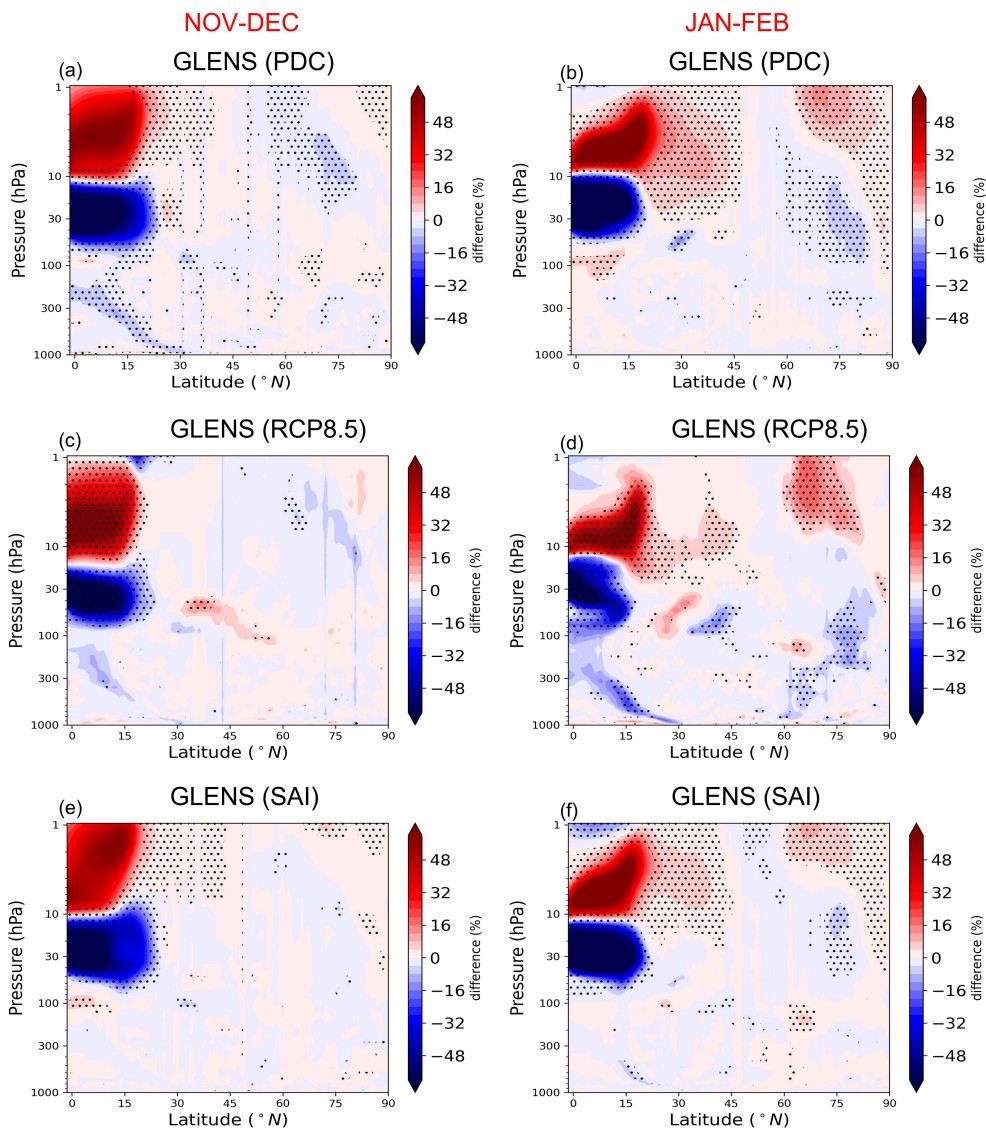

**Figure 8.** Latitude-pressure cross section of the QBOe-QBOw composite differences in percentages of the favorable propagation condition for the resolved large-scale waves (based on the refractive index of Rossby waves). The stippled areas indicate regions where the changes of the quantity are statistically significant at 95% level according to the t test.





Because the QBO modulation of upward propagating planetary waves changes from early winter to late winter and also because of lack of persistence in the wave response, we suggest that the critical line mechanism is not the sole mechanism for changes in the HT relationship in the GLENS simulations. Similar to our results, previous researchers (Dunkerton and Baldwin , 1991; Ruzmaikin et al. , 2005; Naoe and Shibata , 2010; Yamashita et al. , 2011) have documented changes of planetary wave amplitude and/or EP flux and EP flux convergence that are inconsistent with what one would expect from the HT relationship.

This is likely because it is not straightforward to predict the influence of the subtropical critical line in reflecting planetary waves, in particular due to the feedback processes that are initiated as a result of the wave forcing (Watson and Gray , 2014).

     Figure 9 shows the QBOe-QBOw composite differences in the zonal-mean meridional stream lines and zonal mean temperature responses. The positive and negative anomalies of the stream lines indicate clockwise and anticlockwise circulation, respectively. The QBO induces a three-vertical-cell structure of the temperature anomalies both at the equator ($10°S - 10°N$)

and in the subtropics ($10°N - 40°N$ in Nov-Dec and $10°N - 50°N$ in Jan-Feb) in all scenarios and periods. This circulation, known as the secondary meridional circulation, is characterized by sinking motion anomalies at the equator ($10°S - 10°N$) in westerly shear zones and a rising motion in easterly shear zones. In the tropics, the temperature has a thermal wind relationship with the vertical shear of zonal wind such that QBO-induced cold and warm temperature anomalies coincide with easterly and westerly shear zones, respectively (Plumb and Bell , 1982; Randel et al. , 1999; Ribera et al. , 2004). The QBO-induced

temperature anomaly changes sign at approximately $\pm 15°$. The temperature anomalies in the subtropics are out-of-phase compared to the equatorial anomalies as they are related to the return branches of the secondary meridional circulation induced by the QBO. The area of the subtropical temperature anomalies is larger than its tropical counterpart, and they often extend to the mid-latitude winter hemisphere (Kinnersley and Tung , 1998). The latitudinal extension of the temperature meridional cells to mid latitudes occur via modulation of the extratropical Rossby waves (Ribera et al. , 2004; Baldwin et al. , 2001; Kinnersley and

Tung , 1998). The temperature anomalies at high-latitudes correspond well to the changes in the mean meridional circulation. In particular, it is found that in QBOe-QBOw composites (Fig. 9), the streamline anomalies are positive, indicating a clockwise circulation and hence sinking in the high-latitudes; thus, under QBOe the climatological sinking is enhanced compared to the QBOw, which is in agreement with the composites of temperature. Figure 9 also shows that the QBO-induced anomalies of the zonal-mean meridional circulation in high-latitudes in Nov-Dec are not significant under RCP8.5 and SAI simulations, while

they are large and statistically significant in the PDC simulation. This clearly explains the weaker temperature responses under the RCP8.5 and SAI simulations compared to the PDC. Therefore, it can be concluded that the weaker clockwise circulation anomalies in the high-latitude stratosphere in the QBOe-QBOw composites under RCP8.5 and SAI scenarios compared to the PDC simulation in Nov-Dec is the primary reason for the weakening of the HT relationship in RCP8.5 and SAI scenarios and the subtropical zero-wind line plays a less important role.

Figures 10 and 11 show the QBOe-QBOw composite differences of the contributions from the resolved Rossby waves and unresolved gravity waves in driving the mean meridional circulation anomalies. In general, the magnitudes of the composite differences are larger for the gravity waves. Particularly in the high latitudes, the composite differences of the zonal mean meridional circulation due to the gravity waves explains well the temperature responses. It is also found that the resolved waves contribute minimally to the QBOe-QBOw composite differences in temperature in low-latitudes. Instead, the contribution from

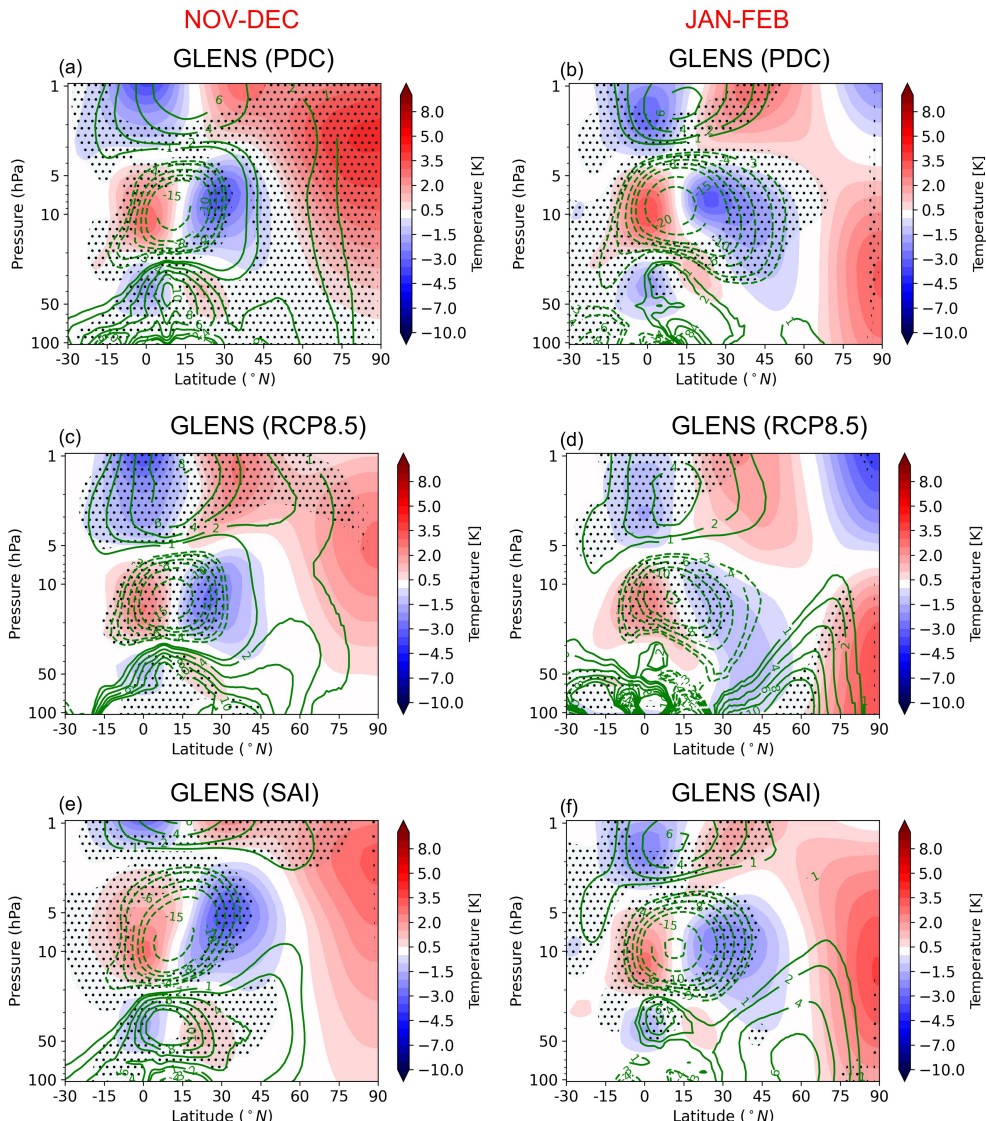

**Figure 9.** Latitude-pressure cross section of the QBO composite differences (QBOe-QBOw) of the zonal-mean meridional stream lines (green contours) and temperature (color shaded). The streamlines are calculated based on equation 7 and hence the contributions from both Rossby and gravity waves are included. The stippled areas indicate regions where the changes of the zonal-mean meridional stream lines are statistically significant at 95% level according to the t test.





the unresolved gravity waves is very robust in these regions. In subtropics and mid-latitude regions, the contribution from the Rossby waves is also considerable. It is worthwhile to mention that gravity waves have a climatologically counterclockwise circulation in region between $45°N - 75°N$ (see Fig. 3c of Okamoto et al. (2011)). Therefore, the positive anomalies in these regions indicate a weakening of the counterclockwise circulation by the gravity wave dissipations under QBOe compared to QBOw. The arched down (downward extension) temperature anomalies towards the tropopause in the midlatitudes is due to

the contribution of both Rossby and gravity waves. In Nov-Dec, the composite differences of the residual circulation due to gravity waves extend to high-latitudes and are statistically significant only for PDC, but that is not the case for the RCP8.5 and SAI simulations and this explains the weaker HT relationship under RCP8.5 and SAI compared to the PDC. In Jan-Feb, the composite differences of QBOe-QBOw of the residual circulation are enhanced in the RCP8.5 and SAI in the mid and high latitudes which is due to the gravity waves contribution.

**5   Discussion and Conclusions**

We examined changes of the HT relationship under a strategic SAI and a high anthropogenic emission scenario pathway (RCP8.5) scenarios, and compared them to the present-day climate (PDC) by using a century-long GLENS project simulations. While the PDC underestimates the HT relationship compared to ERA5 in Jan-Feb, a stronger HT relationship is found in Nov-Dec. Nevertheless, the pattern of the HT relationship is reasonably represented in the PDC, suggesting that the model

configuration is suitable for investigation of the possible future changes. The composite differences (QBOe-QBOw) of zonal wind and temperature show that the HT relationship weakens under both RCP8.5 and SAI scenarios in Nov-Dec, albeit the HT relationship remains closer to PDC under SAI than under RCP8.5. In Jan-Feb, the HT relationship does not change considerable in response to either RCP8.5 or SAI scenarios compared to PDC. While the different phases of the equatorial QBO cause a statistically significant temperature changes in the high-latitude stratosphere under PDC, the responses are not statistically

significant in the RCP8.5 and SAI scenarios.

The central anomaly of QBO at 35 hPa is weaker and its vertical and meridional extents are limited under RCP8.5 compared to PDC. This is also reflected on the location of zero-wind line (critical level) where the distance between the location of zero-wind line is shorter for the different QBO phases under RCP8.5 (particularly in Jan-Feb) compared to SAI and PDC. However, the QBO amplitude under SAI remains comparable to PDC. A weaker amplitude of the QBO under RCP8.5 compared to

PDC and SAI also weakly influences the effective waveguide of Rossby waves under different phases of QBO, such that the wave-mean flow interactions largely differ under RCP8.5 compared to PDC and SAI, which resemble each other.

In general, the temperature responses at high-latitudes corresponds well to the changes in the mean meridional circulation. Our results also show that in Nov-Dec the QBO-induced anomalies of the mean meridional circulation at high-latitudes are not statistically significant under the RCP8.5 and SAI scenarios, while they are statistically significant under PDC. Further

analysis reveals that the gravity wave contribution in composite differences of QBOe-QBOw in Nov-Dec are statistically significant under PDC but this is not the case in RCP8.5 and SAI scenarios. This explains the weaker HT relationship under RCP8.5 and SAI compared to PDC in Nov-Dec. In Jan-Feb, while the composite differences of QBOe-QBOw of the residual

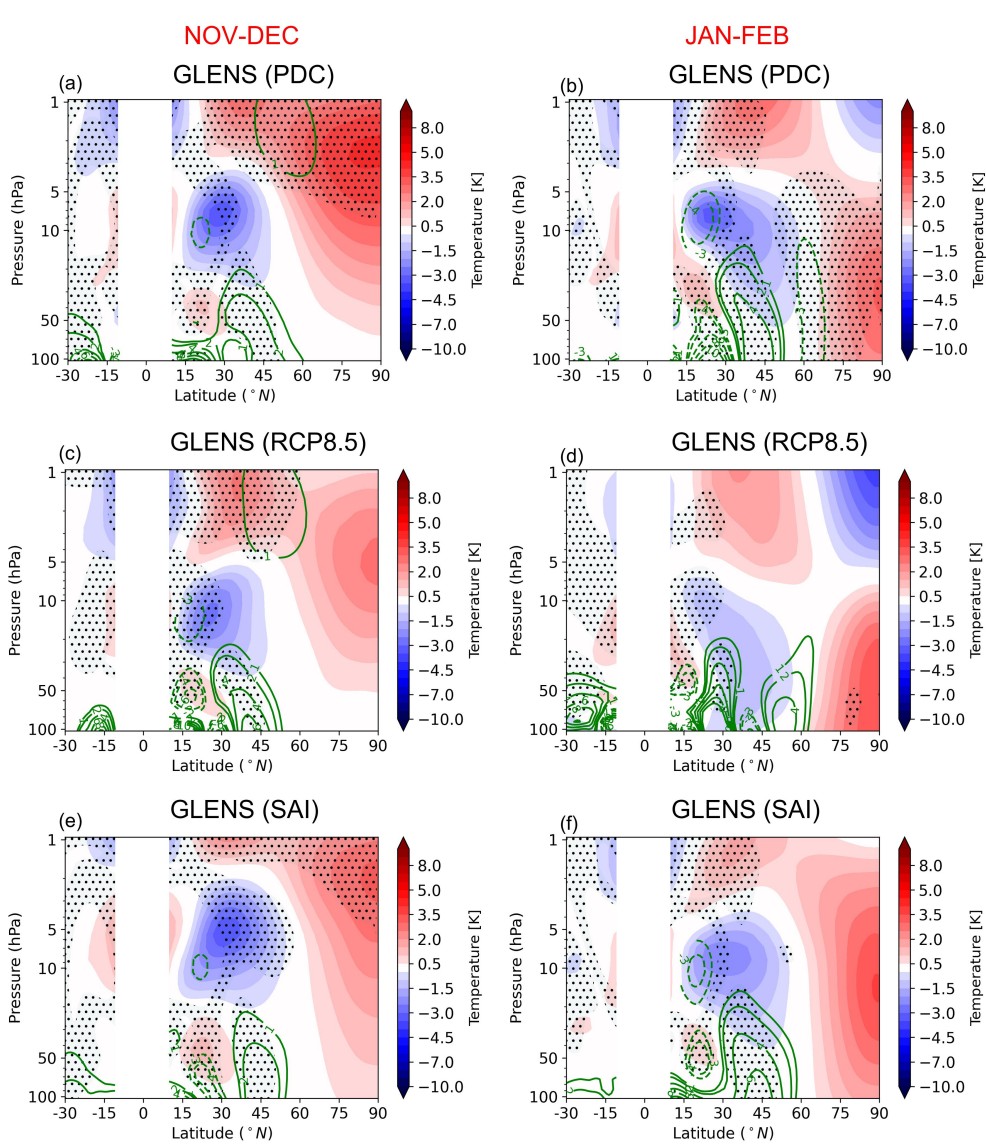

**Figure 10.** The same as Fig. 9 but for the contribution from the resolved Rossby waves.



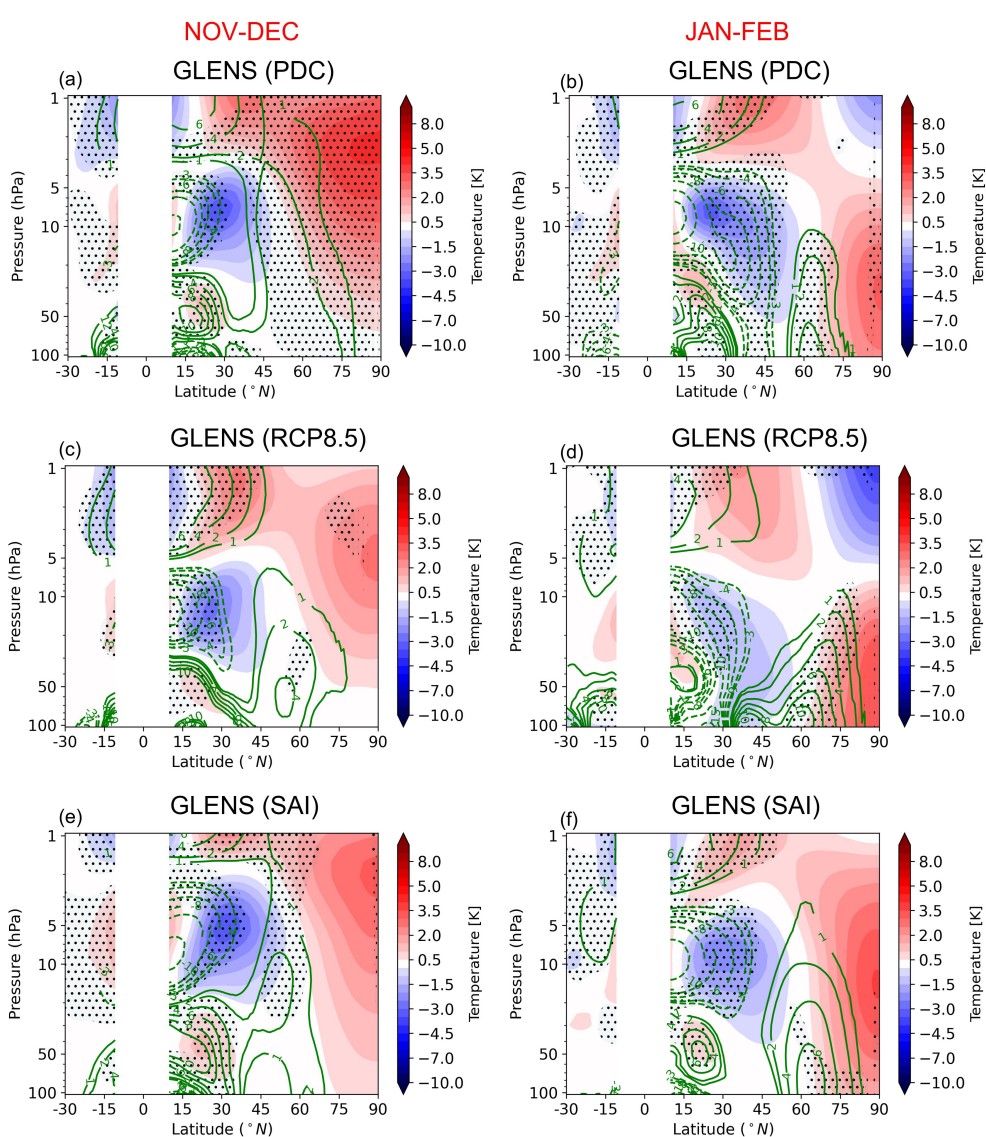

**Figure 11.** The same as Fig. 9 but for the contribution from the unresolved gravity waves.





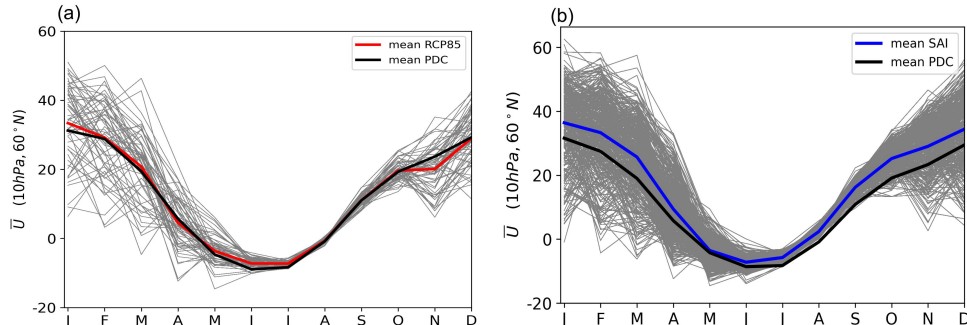

**Figure 12.** Monthly mean evolution of the zonal mean zonal wind at 10 hPa, $60°N$ for the RCP8.5 (a) and SAI (b) simulations. The zonal wind for each year for the RCP8.5 and SAI scenarios are shown by the gray lines and the multi-ensemble multi-year averages are shown by the red and blue lines for the RCP8.5 and SAI, respectively. The black line indicate the multi-ensemble multi-year average of PDC simulation.

circulation are stronger in the RCP8.5 and SAI compared to the PDC (particularly in the mid and high latitudes), however, the strength of the HT relationship does not considerable change under either scenarios compared to the PDC. Therefore, we

seek an alternative explanation for the unchanged HT relationship in Jan-Feb under both RCP8.5 and SAI scenarios. Richter et al. (2018) reported a reduction in the future frequency of sudden stratospheric warmings (SSWs) in both RCP8.5 and SAI scenarios. The frequency of SSWs between 1975 and 2015 was 0.7 per year and it decreased to 0.36 and 0.28 events per year in RCP8.5 and SAI scenarios, respectively, for the period 2060-2099. They also reported that the polar vortex variability does not change significantly under the RCP8.5 global warming scenario compared to the present-day climate, but the vortex became

more stable under the SAI scenario. However, their results were based on a single realization of the RCP8.5 and SAI scenarios. Figure 12 shows the monthly mean evolution of the zonal mean zonal wind at 10 hPa, $60°N$ for both RCP8.5 and SAI scenarios based on all ensemble members of GLENS (20 members for the PDC and SAI and 3 members for the RCP8.5). The average changes in the zonal mean zonal wind at 10 hPa, $60°N$ for the cold season (Oct-Mar) is 0.11 m/s for the RCP8.5 scenario and 5.75 m/s for the SAI scenario. Here we suggest that the unchanged HT relationship under the SAI scenario in Jan-Feb might

be related to the more stable polar vortex (and hence colder polar stratosphere) under SAI compared to the PDC.

Although it is difficult to fully disentangle the contributing factors in the changes of the HT relationship under different climate change scenarios, Table 1 and Table 2 summarize the relative roles of different physical mechanisms in influencing the changes of the HT relationship under the RCP8.5 and SAI scenarios compared to the PDC. The changes in the HT relationship under the RCP8.5 might be explained by the weaker QBO amplitudes (e.g. the weaker QBO amplitudes have weaker potential

to influence the high-latitudes). However, under the SAI scenario the QBO amplitudes do not significantly differ compared to the PDC, but the polar vortex accelerates and we attribute the weakening of the HT relationship under SAI scenario to the more stable polar vortex. Another relevant point in the interpretation of the QBO composite differences of the zonal-mean meridional stream lines due to the resolved and unresolved waves is the so-called compensation mechanism, whereby the





**Table 1.** The relative composite differences (QBOe-QBOw) of different physical mechanisms influencing the strength of the HT relationship in Nov-Dec. The weaker and stronger mechanisms are relative to the PDC. A blank indicates insignificant changes compared to the PDC.

|        | QBO amplitude | polar vortex strength | meridional circulation | HT relationship           |
| ------ | ------------- | --------------------- | ---------------------- | ------------------------- |
| RCP8.5 | weaker        | -                     | weaker                 | weaker (near elimination) |
| SAI    | -             | stronger              | weaker                 | weaker                    |

**Table 2.** The same as Table 1 but for the Jan-Feb period.

|        | QBO amplitude | polar vortex strength | meridional circulation | HT relationship   |
| ------ | ------------- | --------------------- | ---------------------- | ----------------- |
| RCP8.5 | weaker        | -                     | stronger               | weaker (slightly) |
| SAI    | -             | stronger              | stronger               | weaker (slightly) |

changes in the gravity wave forcing are compensated by the resolved wave driving of opposite sign (Cohen et al. , 2001; Sigmond and Shepherd , 2014; Garcia et al. , 2017; Eichinger et al. , 2020).

Finally, we would like to mention that the results presented here are based on a relatively coarse vertical resolution simulations (70L model) which has a somewhat deficient QBO. It is important to investigate the robustness of these findings with a higher vertical resolution version of CESM1(WACCM) to gain additional confidence in our results. In addition, examining the changes in the HT relationship under the SAI scenario using other Earth system models would be useful due to uncertainties inherent in complex Earth system models.

*Data availability.* The NCAR's GLENS simulation datasets can be downloaded from here: https://www.cesm.ucar.edu/projects/community-projects/GLENS/.

*Author contributions.* The figures are produced by Khalil Karami. Simone Tilmes lead and Jadwiga Richter contributed to the GLENS simulation. All co-authors contributed to writing and interpreting the results.

*Competing interests.* All authors declare that they have no competing interests.

*Acknowledgements.* This material is based upon work supported by the National Center for Atmospheric Research, which is a major facility sponsored by the National Science Foundation under Cooperative Agreement no. 1852977. The Community Earth System Model (CESM)



project is supported primarily by the National Science Foundation. This study is funded by the Deutsche Forschungsgemeinschaft (DFG) under grant number JA 836/47-1 and within the Transregional Collaborative Research Centre SFB/TRR 172 (Project-ID 268020496), sub-
project D01. This work used resources of the Deutsches Klimarechenzentrum (DKRZ) granted by its Scientific Steering Committee (WLA) under project ID bb1238.



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
