# Peer review of "The Holton-Tan mechanism under stratospheric aerosol intervention"

_Atmospheric Chemistry and Physics, 2022_

## Author Comment (AC1)

Responses to the first reviewer's comments

We sincerely appreciate your effort and time in reviewing our manuscript as well as constructive comments/suggestions. We have revise the manuscript in the light of your suggestions/comments. The following is also the point-by-point response to all the comments. Please note that the reviewer's comments are written in bold font, and our responses are written using a regular font.

Major comments:

1)The authors argue that the HT effect weakens both in the SAI and RCP8.5 scenarios in both early and mid-winter. The evidence provided does not support this conclusion with regards to the mid-winter, clearly contradicting e.g. line 6 of the abstract. The slopes in Figure 2 for Jan/Feb are indistinguishable for the three model runs (but please add error bars for the slopes!). Figure 3 and 4 also demonstrate (to my eye) no difference among the three for Jan/Feb. Interestingly, the tropospheric jet shift and presumably surface impact in Figure 3 is actually stronger for RCP8.5 than for PDC! (a finding which agrees with Rao et al 2020). This is actually acknowledged near line 212, however there appears to be a lack of consistency within the paper as to this conclusion. Line 241/242 just adds to the confusion. Line 323/324 adds yet another interpretation of the results which again differs from any previously offered.

To be constructive, please add stippling for statistical significance of the difference between SAI vs PDC and RCP8.5 vs PDC on Figure 5 and 6. Next, revise the text in the aforementioned locations to make the text and figures internally consistent.

 $\checkmark$  Our response: In the revised version, we added the 95% confidence intervals as shaded areas. The stippling for statistical significance of the difference between SAI vs PDC and RCP8.5 vs PDC on Fig. 5 and Fig. 6 are also added.

We also made the following revisions to make the text and figures consistent: The abstract is rewritten: "... Our results from an Earth system model indicate that, under both global warming (based on RCP8.5 emission scenario) and SAI scenarios, the HT relationship weakens in early winter (Nov-Dec), although it is closer to PDC under SAI than under the RCP8.5 scenario. In contrast, the HT relationship in mid-late winter period (Jan-Feb) does not change considerably in response to either RCP8.5 or SAI scenarios compared to PDC. ... ".

Line 241/242 in the previous submission is also revised. In the revised version we have: "The weakened HT relationship in Nov-Dec under the SAI scenario cannot be attributed to the changes in the QBO anomalies in the equatorial region, as it largely resembles the QBO anomalies under the PDC simulation.". The above-mentioned changes are consistent with the text in Line 323/324 (initial submission) as well as Table 2.

Investigating the surface influence of the Holton-Tan ought to be carried out

in a model with high vertical resolution. The CMIP6 (standard, 70L) version of WACCM is not useful for such study. In fact, the authors looked at this in the early version of the manuscript but such connection was not found (even in the control run). There is a 110L WACCM6 run through 2100 that was carried out for the QBOi intercomparison in NCAR. But unfortunately that is not a geoengineering run. Future simulations of geoengineering at NCAR should take this into account.

We added the following text to the first paragraph of section 5 (Discussion and Conclusion): "It is worthwhile to mention that the HT relationship in this study only refers to the teleconnection between the QBO and the Arctic stratospheric polar vortex and other aspect of QBO influence such as surface impacts are not studied in this paper. One reason is that investigating the surface influence of the QBO via the stratospheric polar vortex path requires a model with high-vertical resolution. The WACCM6 with a relatively coarse vertical resolution of 70 layers between surface and model top at 140 km is not useful for such investigation."

2. For the index of refraction analysis, is N (Brunt-Vaisala frequency) allowed to vary in the vertical? The original Matsuno paper explicitly holds N constant, however eq 13 in this paper includes it inside the derivative. If you allow for N to spatially vary, you may as well use the Weinberger et al 2021 definition of the index of refraction which includes more physical processes without being any less physically inconsistent.

 $\checkmark$  Our response: N (Brunt-Vaisala frequency) in equation 12 and 13 is not constant but it varies spatially (Sun and Li, 2012; Sun et al., 2014).

3. The stippling on figure 4 seems counter-intuitive, and even possibly incorrect. Is the deep tropical temperature anomaly associated with the QBO really not significant for a composite based on QBO winds? This temperature response should be highly consistent from one event to the other, and should be very robust. The top row looks reasonable, but not the rest.

 $\checkmark$  Our response: Thanks for identifying and mentioning this issue. There was a coding error (in the plot, not in the calculation of significance), which is corrected in the revised version.

4. I disagree with the interpretation offered in the first paragraph of section 4 where it is claimed the results agree with the HT mechanism. The  $F_y$  feature focused upon is well above the region of easterlies and is instead near 35-40km where westerlies prevail. Rather, this seems to be more consistent with the mechanism of Garfinkel et al 2012 as it occurs where the MMC of the QBO is strong, though it is hard to see the effect of Garfinkel et al 2012 in the index of refraction figures. However, figure 7 and figure 8 do not share the same y axis and labeling, so it is difficult to reach any conclusions as to whether  $n^2$  changes and EPF changes are consistent in a given region. Please revise the axes of figures 7 and 8 to make them consistent, and then provide a more detailed analyses

**of possible consistencies between the diagnostics.**

Thanks for mentioning this. We rewrote this paragraph as: "Figure 7 shows the composite QBOe-QBOw differences of the EP flux, its divergence, the zonal mean zonal wind response, and the location of the zero wind line for the different QBO phases. We found that under QBOe the climatologically equatorward component of the EP flux is weaker than under QBOw, and hence the composite differences show that under different climate change scenarios the resolved waves are always directed towards the high latitudes in the middle and upper stratosphere. At first glance, this might seem consistent with the traditional HT hypothesis (Holton and Tan, 1980) that under QBOe, the quasi-stationary waves cannot propagate in easterly winds and therefore the effective waveguide for the planetary waves is narrower compared to the QBOw case. However, these regions are generally well above the region of easterlies and are above 30 km where westerlies prevail. This is shown in Fig. 8 as the positive anomaly of the favorable propagation condition for Rossby waves between 10-1 hPa which is a pattern that is generally found in all scenarios. While the zero-wind line is on the winter side of the equator (20-30 km) for QBOe under the PDC and SAI scenarios, the distance between the location of the zero-wind line is smaller for the different QBO phases under RCP8.5 (particularly in Jan-Feb). This is also reflected in Fig. 8 where the anomalies of the favorable propagation condition for Rossby waves are meridionally limited and also weaker under RCP8.5 compared to the SAI and PDC simulations particularly in Jan-Feb. This is also expected as the zonal mean zonal wind anomalies (Fig. 3 shows similar patterns. While the weaker equatorward EP flux component in QBOe-QBOw composites are a persistent feature in all scenarios and periods, the vertical EP flux component shows inconsistent responses. In general, in early winter (Nov-Dec) an enhanced upward EP flux is found in the high-latitude stratosphere in all scenarios. On the other hand, in mid-late winter (Jan-Feb) a reduction of the upward EP flux is found in all scenarios. Although the polar vortex is modulated as expected (see Fig 3 and Fig 4), the wave propagation does not follow the HT mechanism, which is in agreement with the study of Naoe and Shibata (2010)and Yamashita et al (2011). This suggests that the effect of the zero-wind line emphasized in the HT mechanism is not so important in the QBO modulation of the wintertime stratospheric polar vortex.

Alternatively, Garfinkel et al (2012), by performing idealized experiments using the WACCM model, reported that the the subtropical critical line modulates the synoptic Rossby wave convergence in the subtropical lower stratosphere and cannot directly influence the large-scale waves in the middle and higher latitudes. Instead, they suggest that the secondary meridional circulation of the QBO, which modifies the zonal-mean wind distribution, affects the refractive index and this influences the wave propagation in the mid and higher latitudes of the upper stratosphere. It is also worthwhile to mention that it is challenging to isolate the exact mechanism(s) by which the QBO influences the vortex in the comprehensive GCMs such as our dataset due to the presence of unrelated variability. For example, variability in sea surface temperature (SSTs) not only influences the tropospheric planetary wave activity (Garfinkel and Hartmann 2008; Hurwitz et al 2011) but also affects the strength of the HT mechanism (Holton and Austin 1991; O'Sullivan and Dunkerton 1994). Although the sign of the refractive index and EP flux in our analysis are somewhat different that those reported by Garfinkel et al (2012) (likely due to the more idealized experiments in Garfinkel et al (2012) with fixed SSTs, land surface and ice and perpetual mid-winter radiative forcing and the lack of interactive chemistry that might mask the HT mechanism (Wei et al., 2007; Garfinkel and Hartmann 2007; Calvo et al., 2009)), nevertheless we will show that the high-latitude modulation of the polar vortex in different scenarios can be explained by the changes in the residual circulation consistent with Garfinkel et al (2012). "

---

## Author Comment (AC2)

Responses to the Second reviewer's comments

We sincerely appreciate your time in reviewing our manuscript as well as your suggestion. Please note that the reviewer's comments are written in bold font, and our responses are written using a regular font.

✓ **The paper studies the sensitivity of the Holton-Tan mechanism to global warming and stratospheric geoengineering. This mechanism plays an essential role in different phenomena, from the climate impact of volcanic eruptions to stratospheric aerosol interventions (SAI). To study these complex dynamic processes associated with the interaction of QBO and polar vortex, the authors used may be the best available modeling tools that can simulate QBO dynamics and frequency realistically. I have little to criticize the paper. It is well-written and logically organized, and the scientific approach and the analysis are sound. The only suggestion that could add to the value of this study could be an explicit link to the tropospheric climate responses, like winter warming in high northern latitudes over the Euroasian Continent.**

✓ our response:

We thank the reviewer for the suggestion. Investigating surface influence of the Holton-Tan ought to be done in a model with high vertical resolution. The CMIP6 (standard, 70L) version of WACCM is not useful for such study. In fact, the authors looked at this in the early version of the manuscript but such connection was not found (even in the control run). There is a 110L WACCM6 run through 2100 that is done for the QBOi at NCAR, which is, however, not a geo-engineering run. Hopefully, the future simulations of geoengineering at NCAR will take this into account.

We added the following text to the first paragraph of section 5 (Discussion and Conclusion): "It is worthwhile to mention that the HT relationship in this study only refers to the teleconnection between the QBO and the Arctic stratospheric polar vortex and other aspects of QBO influence such as surface impacts are not studied in this paper. One reason is that investigating the surface influence of the QBO via the stratospheric polar vortex path requires a model with high vertical resolution. The WACCM6 with a relatively coarse vertical resolution of 70 layers between surface and model top at 140 km is not useful for such investigation."

---

## Author Response (AR2)

Responses to the first reviewer's comments

We appreciate your effort and time in reviewing our manuscript as well as minor comments. We have revise the manuscript, accordingly. Please note that the reviewer's comments are written in bold font, and our responses are written using a regular font.

**The present version of the manuscript is much clearer and self-consistent. There are still some minor revisions needed, however the overall message is ok**

**1. In my initial response, I noted that the original Matsuno paper explicitly holds N constant, however this paper allows it to vary in the vertical. The reviewers cited two other studies that also make this assumption in their response. I realize that this paper isn't the first paper to grossly violate the conditions under which Matsuno derived his index of refraction. My point is that if you are already taking the theory well outside of its original application, you may as well use the Weinberger et al 2021 definition of the index of refraction which includes more physical processes without being any less physically inconsistent.**

**I suspect the results won't be very much different, however some discussion of this point is needed.**

✓ Our response:

Thanks for the suggestion. In the form of the refractive index used in equation 12, $N$ varies in the vertical direction and hence is sensitive to its changes near the tropopause and therefore this form of refractive index is suitable for the current study.

As suggested, the following text is added to the manuscript :

It is worthwhile to mention that the index of refraction has different forms. Weinberger (2021) has evaluated the ability of four different versions of the index of refraction to capture the upward wave propagation from the troposphere to the stratosphere. In particular, they show that the vertical gradients in buoyancy frequency near the tropopause is critical for understanding the upward wave propagation from the troposphere. In the form of the refractive index used in equation 12 the buoyancy frequency varies in the vertical direction and hence is sensitive to its changes near the tropopause and therefore is suitable for the current study.

**Minor comments:**

**line 87: Whether CMIP5/6 underestimate the HT effect or not was discussed by Rao et al 2020. They found that if one accounts for the lack of downward propagation to the lowermost stratosphere and the particular QBO regimes simulated by each model, there isn't an underestimate. However there is an underestimate if one uses a simple QBO definition based on fixed pressure levels.**

**2. the stippling on figure 4 and 6 still look incorrect to me. Particularly noticeable are the bottom row.**

✓ Our response:

To make sure that the calculations including the stippling points are corrected, we checked the codes. They are correct.

The text is modified as the reviewer suggested:

"Most Coupled Model Intercomparison Project 6 (CMIP6) models underestimate the HT relationship in the present-day climate (Elsbury2021) although this depends on the QBO definition and the particular QBO regimes simulated by each model (Rao2020b)."

**line 179 "tropics and lower latitudes": should "lower" be replaced by "higher"**

✓ Our response:

Thanks for identifying it. It is corrected in the revised version.